# Opening and Reusing Transparent Peer Reviews with Automatic Article Annotation

**Afshin Sadeghi [1,2,*]**, **Sarven Capadisli [1]**, **Johannes Wilm [3]** and **Christoph Lange [1,2]**
and **Philipp Mayr [3]**

[1]  Institute for Applied Computer Science, University of Bonn, 53012 Bonn, Germany; info@csarven.ca (S.C.); langec@cs.uni-bonn.de (C.L.)
[2]  Fraunhofer Institute f. Intelligent Analysis and Information Systems IAIS, 53757 Sankt Augustin, Germany
[3]  GESIS—Leibniz Institute for the Social Sciences, 68159 Mannheim, Germany; mail@johanneswilm.org (J.W.); philipp.mayr@gesis.org (P.M.)
*  Correspondence: i@afshn.com

**Abstract:** An increasing number of scientific publications are created in open and transparent peer review models: a submission is published first, and then reviewers are invited, or a submission is reviewed in a closed environment but then these reviews are published with the final article, or combinations of these. Reasons for open peer review include giving better credit to reviewers, and enabling readers to better appraise the quality of a publication. In most cases, the full, unstructured text of an open review is published next to the full, unstructured text of the article reviewed. This approach prevents human readers from getting a quick impression of the quality of *parts* of an article, and it does not easily support secondary exploitation, e.g., for scientometrics on reviews. While document formats have been proposed for publishing structured articles including reviews, integrated tool support for entire open peer review workflows resulting in such documents is still scarce. We present **AR-Annotator**, the Automatic **Article and Review Annotator** which employs a semantic information model of an article and its reviews, using semantic markup and unique identifiers for all entities of interest. The fine-grained article structure is not only exposed to authors and reviewers but also preserved in the published version. We publish articles and their reviews in a Linked Data representation and thus maximise their reusability by third party applications. We demonstrate this reusability by running quality-related queries against the structured representation of articles and their reviews.

**Keywords:** automatic semantic annotation; open peer review; knowledge extraction; open science; electronic publishing on the Web

## 1. Introduction

The scientific community expects that the quality of published articles is controlled in a review process. When an author has finished writing an article, it is typically reviewed by two or more independent experts. Successfully passing the review process is a prerequisite for an article to be published. Besides generally recommending whether or not the article in its current shape has a sufficient quality to be published, reviewers give specific feedback on parts of the article, often including detailed comments suggesting improvements. When authors are notified of reviews, they take them into account in their next revision of the article. Traditionally, reviews have mostly been confidential, whereas today there is a strong movement towards *open* peer reviews. We discuss the benefits of open peer reviews in Section 1.1, state the lack of tool support for publishing open reviews as a problem in Sections 1.2 and 2, present concrete use cases for a system that publishes peer-reviewed



articles on the Web appealing to human and machine clients, and derive precise requirements from these use cases. Section 3 discusses related approaches specifically from the perspective of how well they support our use cases and how well they address our derived requirements. Section 4 presents our implementation, which supports the given use cases and addresses the given requirements. It comprises a pipeline in which Fidus Writer, Open Journal Systems (OJS), and dokieli are connected by the core of AR-Annotator, which translates a document that is commented on resulting from the combined usage of Fidus Writer and OJS to dokieli's HTML+RDFa article format, preserving the original structure of the document but further enriching it with extracted semantics. Section 5 demonstrates the usefulness of attaching reviews to documents in a study.

### 1.1. Open Peer Review

In traditional workflows, reviews are confidential. The programme committee chair of a conference or the editor of a journal initially receives the reviews and passes them on to the authors. The identity of the reviewers is typically not disclosed to the authors ("single-blind review"); in some communities, the reviewers do not know the identities of the authors either ("double-blind review"). Nowadays, there is a strong movement towards *open peer reviews* in many communities [1]. One common variant starts with a traditional closed process, after whose end the reviews (including the identities of the reviewers) are published (sometimes with the possibility for reviewers to opt out); in another variant, the unreviewed article is published first and then reviewed. The open peer review policy employed by the *Semantic Web Journal*, for example, aims to "minimize positive and negative bias in the reviewing" [2]. *BMJ Open* [3] is another journal that publishes all the reviews alongside the original article, as well as the authors' responses to the reviews. Another pioneer in open scholarly publishing is Data Science Journal [4], which not only discloses the reviews of articles but also publishes the responses of authors to the comments and decisions of the reviewers. Some studies suggest a positive effect of this openness on the quality of reviews [5,6]. Other works argue in favor of adding more transparency and self-regulation to the peer review process when opening reviews up [7–9]. Further reasons for open peer review include giving better credit to the work done by the reviewers, as well as enabling readers to better appraise the quality of a publication.

### 1.2. Problem Statement: Lack of Tool Support

Review processes are technically well supported by submission and review management systems such as EasyChair [10] for conferences, or Open Journal Systems [11] for journals. However, these systems focus on facilitating the assignment of reviewers to submissions and on automating the sending of notifications by email. Besides requesting the client to enter some global numeric scores, they treat the actual review as an unstructured block of text. When such a review is entered into the system, any explicit information on the connection of reviewers' comments to parts of the article is lost. Readers of such reviews have to read the review in full and switch back and forth between the review and the article in order to understand the connection. This loss of context most severely affects authors, aiming at understanding where they should improve their article, but it may also affect readers who are interested in gaining a deep understanding of an article published openly along with its reviews. Even if reviewers point to precise text locations, such as "the caption of Figure 1" or "the last sentence on page 7" (which is tedious without software support), it takes readers (including the authors as a special case) time, and it is error-prone to interpret these references manually. State-of-the-art tools for reading and writing articles support comments anchored to precise ranges of text, thus eliminating any ambiguity regarding what part of the article a reviewer's comment refers to. Submission and review management systems are often configured to accept the upload of attachments, which could be a version of the article annotated with such comments. If such facilities were used more frequently, such reviews published openly would be easier to consume for human readers than traditional ones—as discussed in Section 5, but their utility would still remain limited from the perspective of *automated* approaches for *secondary exploitation* of reviews, e.g.,

for giving credit to reviewers or for appraising the quality of an article. For example, it would still require human intelligence to understand whether the authors' observations from the evaluation of a system were accepted by the reviewers or discussed controversially. Document formats such as HTML+RDFa [12], combined with suitable vocabularies such as the SPAR (Semantic Publishing and Referencing) ontologies [13] for making the structure of the article explicit, or the Web Annotation Vocabulary [14] for making the provenance of comments and their connection to the text explicit, support the publication of articles and reviews in a way that appeals both to human readers and to machine services. However, document editors as well as submission and review management systems addressing end users without a web and knowledge engineering background have so far not supported such formats: neither have document editors supported an explicit representation of the structure of an article (such as "this is the evaluation section"), nor have review management systems supported human as well as machine friendly publication of reviewers' comments attached to parts of articles in a fine-grained way.

The contributions of this work are as follows:

- A pipeline, named AR-Annotator (Article and Review Annotator), to enrich articles and reviews with RDFa data, immediately preparing them for efficient analysis by SPARQL queries, and furthermore providing a way to allow them to be added to the LOD [1] Cloud and scholarly communication knowledge graphs such as SCM-KG knowledge graph [15].
- We support the repeatability and reproducibility of our approach and the empirical evaluation results by providing the source code of the article enriching module and the evaluation procedure under Apache license [16].

Section 6 concludes with an outlook to future work and a sketch of the mid-term impact that a wide adoption of AR-Annotator could make on scholarly communication.

## 2. Use Cases and Derived Requirements

Section 2.1 presents concrete use cases for a system that publishes peer-reviewed articles on the Web appealing to human and machine clients. Section 2.2 derives precise requirements from these use cases. Section 5 presents a plausibility test that justifies the specific requirement to attach reviewers' comments to parts of articles in a fine-grained way by proving the usefulness of doing so.

### 2.1. Use Cases

We first outline use cases our approach aims to support. Overall, open science aims at supporting the reuse of research output and making transparent the advancement of human knowledge using the Web. We aim at pushing this approach by better utilizing the interactivity potential of Web technology to realize an integrated authoring, reviewing and publishing workflow.

### 2.1.1. Soliciting Additional Reviews

We aim at publishing articles together with their reviews, in the first step focusing on reviews coming from a closed workflow. Nevertheless, we would like the published article to attract further comments from new reviewers in the community, even after acceptance. This is helpful in settings when a sufficient number of reviews cannot be obtained by traditional means, or, more generally, when the authors but also readers welcome additional feedback.

### 2.1.2. Automated Analysis of Reviews

Several scenarios could benefit from an automated analysis of reviews by repeatable queries against the review comments and the structure of the article. Not only would this help the primary

---

[1]　LOD here means linked open data.

recipients of reviews, i.e., the authors, to faster comprehend the reviewers' feedback when revising their article, by answering queries such as "What range of text in the Introduction section have received the most attention from the reviewers?", it would also help editors of a journal, or chairs or senior PC members of a conference, to realize commonalities and differences between the comments of the original reviewers, e.g., "What sections in the article received greatest number of negative comments by reviewers?". Finally, such queries could give rise to new scientometrics based on secondary exploitation of reviews, e.g., "To what extent do reviewers disagree with the 'Methods' section?".

### 2.1.3. Sharing and Reusing Articles and Reviews

To support further reuse of articles and reviews, e.g., by sharing them in social networks or by enabling precise citations, we would like each of their structural components (sections, figures, references, comments, etc.), including each reviewer's comment, to be identifiable—and thus linkable—in a globally unique way and to carry its own local metadata.

### 2.1.4. Multi-Device Accessibility of Reviews

By publishing reviews using Web technology, we aim at increasing their accessibility by making them readable across devices, including devices with different screen sizes, as well as different output modalities including text and speech, and by making their appearance, e.g., font size, adaptable to the requirements of the reader.

### 2.1.5. Independent Quality (Re-)Assessment

Making explicit the structure of an article can support the review process by giving easier access to the resources underlying an article, such as research datasets and software artifacts, thus facilitating the reproducibility of research and reducing the effort of an independent (re-)assessment of an article's quality.

### 2.2. Requirements

From the use cases outlined so far, we derived the following technical requirements for a system supporting the workflow of authoring, reviewing and publishing (in this order):

1.  It should be possible to create initial reviews for an article in a closed submission and review management system.
2.  The support for comments and annotations is the most requested feature from the reviewing tools [17]. Reviewers should be enabled to attach comments in a fine-grained way to precise structural elements of an article, and these attachments should be preserved in the published document.
3.  The structure of all parts of an article should be exposed explicitly; each structural component of an article should have a globally unique identifier.
4.  Not only should the structure of the article from the authoring environment be preserved, but, where possible, structural information implicitly hidden in unstructured text should be extracted, and the text of the published article should be enriched with an explicit representation of such structures.
5.  In line with the FAIR principles of making data Findable, Accessible, Interoperable and Reusable [18], our system should publish reviews in a standard web format, which humans can consume with a browser, but which can also carry semantic metadata enabling queries and other automated analyses.
6.  For compatibility with post-publication peer review processes, it should be possible to add further reviews to documents once it has been published.

Besides legal openness in terms of open access, these requirements enable openness in a technical sense.

Note that Requirement 4 applies to the publishing of articles in general, not just to peer-reviewed articles. However, making explicit the structure of articles helps to understand the context in which reviewers attached a comment.

As an additional requirement, we aim at compatibility with traditional workflows. We do not aim at entirely disrupting workflows. It should remain possible to subsequently open up articles and reviews created in a traditional, closed workflow.

## 3. Related Work

We first review related work focusing on tool support for open peer review, and then studies that have targeted the annotation of articles.

### 3.1. Tool Support for Open Peer Review

With an increasing adoption of open peer review, as explained in Section 1.1, the situation has improved over 2012, when Kriegeskorte argued that the leading review systems hardly published general comments about articles [19]. However, none of the open peer review journals we have reviewed provide technical support for publishing reviews with fine-grained comments attached to parts of articles. To the best of our knowledge, state-of-the-art online open publishing systems do not offer advanced facilities for publishing articles and reviews in a way that make them easy to consume for humans, let alone machines. For example, the client-server *Open Journal Systems* [11] (OJS) supports a sophisticated reviewing workflow management featuring different user roles and access permissions as well as the publication of reviews once they have been finished, but only as a block of text in the static PDF format.

To enable open peer review with fine-grained comments in a "publish first" setting, one could simply employ collaborative web-based authoring tools with commenting facilities, such as Google Docs [20], which targets non-technical users. However, this approach does not support export of reviews and it is not integrated with any review workflow. In our previous work, we have integrated Fidus Writer [21], an academically oriented alternative to Google Docs, with OJS to combine collaborative authoring and a sophisticated management of a traditional peer review workflow [22]—but this combination does not support the publication of articles annotated with fine-grained reviewers' comments because the articles in a Fidus Writer installation can only be viewed by other users registered there, while online publishing is only possible by exporting, e.g., to HTML, which loses the comments.

*dokieli* [23], a web-based tool for editing and publishing articles in human as well as machine-friendly semantic HTML+RDFa markup developed by the second author, supports commenting published articles in a decentralised architecture [24]. However, it does not provide dedicated support for the scholarly reviewing workflow *before* publishing.

### 3.2. Tools for Annotating Articles

Furthermore, dokieli enables linking to structural parts of articles and to comments, and it supports *manual* enrichment of articles with semantic annotations. Other web-based tools that support authors with annotation facilities but not specifically tailored for scientific articles include Loomp [25], a general-purpose annotation framework targeted at journalists, which does not perform automatic annotation either, and the RDFaCE [26] RDFa content editor, which supports automated annotation of news articles.

Several systems support automatic annotation of the content of articles from specific fields of science with ontologies for the respective domains. For example, DOMEO [27] and BioC [28] use text mining services to analyze the bodies of articles and to relate them to biomedical ontologies. The more recent AnnoSys supports generic annotations of articles [29]. However, it supports Web publishing

and annotates in a local, fine-grained way, which would facilitate consumption. It publishes the annotations as separate XML documents in contrast to dokieli, which embeds them as RDFa. The ACM Article Content Miner [30] uses the SPAR ontologies to extract classes that describe the structure of articles. Although mining for SPAR-related keywords allows extracting author affiliations, references, etc. from articles, ACM keeps the extracted information separated from the articles.

## 4. Implementation

This section presents AR-Annotator (Article and Review Annotator), and the implementation of our proposed approach, beginning with the overall architecture of a pipeline comprising the steps of authoring, reviewing and publishing in Section 4.1. Then, in Section 4.2, a description of the previously existing components of which our implementation of the pipeline is composed—the Fidus Writer document editor for authoring and commenting, Open Journal Systems (OJS) for managing the review process, and dokieli for publishing the reviewed article on the Web, and, finally, in Section 4.3, the core of AR-Annotator, which translates a commented document resulting from the combined usage of Fidus Writer and OJS to dokieli's HTML+RDFa format, preserving its original structure and further enriching it by an explicit representation of additional structural aspects.

### 4.1. Architecture

Figure 1 provides a conceptual overview of the pipeline supported by AR-Annotator for authoring, reviewing and publishing scholarly articles. The AR-Annotator module is located between a tool for writing articles and the Web as the final publication infrastructure. It receives an article as a document with formatted text, tables, images and reviews belonging to it and exports a semantically enriched article in semantic Web-compatible markup, which is ready for human and machine consumption.

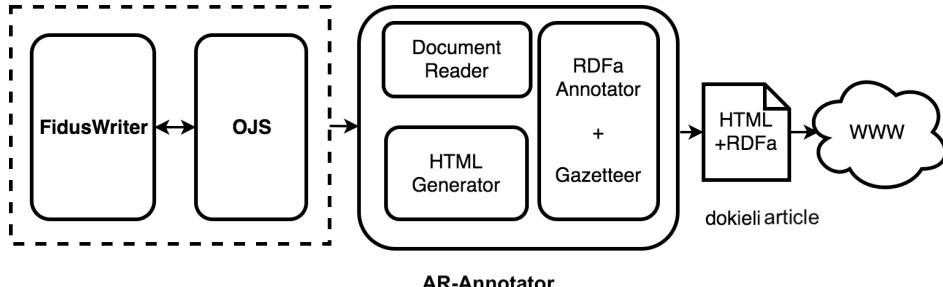

**Figure 1.** A pipeline for scholarly article authoring, reviewing and publishing facilitated by an AR-Annotator.

### 4.2. Integration of Reused Components: Fidus Writer, OJS and dokieli

This subsection presents the existing components that complete the AR-Annotator pipeline and explains why we chose them. We argue how transforming an article written in Fidus Writer to dokieli's HTML+RDFa format enables open scientific publishing and reviewing. Going beyond the discussion of the previous state of these tools in Section 3, we discuss how the AR-Annotator core fills the gap between them by implementing this transformation step.

The initial, closed review is conducted via a centrally hosted combination of Fidus Writer and Open Journal Systems (OJS) [22], which allows authors, reviewers and journal editors to cooperate seamlessly across the boundary of the two systems after only registering once. This integration frees the authors from having to export their article from Fidus Writer, e.g., to PDF, and to upload that copy to OJS, and from once more having to download the article once it has been reviewed. Instead, the reviewers work in the original collaborative Fidus Writer document. After the end of the review process, their comments become visible to the authors right in place.

As discussed in Section 3.1, the Fidus Writer/OJS combination does not support high-quality online publication of articles that contain reviews. On the other hand, dokieli documents can be

displayed in a visually appealing way thanks to supporting the styles of publishers (currently LNCS and ACM authoring styles). They may include comments as well as machine-readable semantics (in RDFa), making explicit the structure of an article. Furthermore, individuals can store their own comments at their personal storage which references a dokieli article or a part within. However, dokieli alone does not support the management of a review process in a way as sophisticated as OJS. Thus, to benefit from the best features of both systems, we combine them by extending Fidus Writer, which has so far only featured a simple HTML export filter as well as LaTeX, ODT and DOCX export filters, all without support for comments, with a facility that exports to the HTML+RDFa format of dokieli, as detailed below in Section 4.3.

The conservative Fidus Writer/OJS approach risks moving publishing to the Web without taking advantage of the opportunity to reform parts of the publishing process that no longer correspond to the current state of the development of the means of communication, ultimately limiting the review process to a small number of academics with privileged access to a central infrastructure. On the other hand, dokieli articles can have their own decentralised "inboxes", i.e., containers in which they can receive notifications, e.g., from reviewers' comments or other annotation activities [31].

Table 1 highlights the respective strengths of the two selected components with regard to criteria corresponding to the requirements that we have derived from our use cases in Section 2. It is obvious that the strengths are complementary and that therefore the integration of both components fully supports the desired workflow.

**Table 1.** Strengths of the selected components w.r.t. our requirements.

| Criterion/Corresponding req. | Fidus Writer | dokieli |
|---|---|---|
| Discoverability of reviews (2, 3, 4, 5) | N/A | Yes (RDFa) |
| Machine-comprehensible representation (3, 4) | N/A | Yes |
| Creation and access to reviews (1) | post publication on article snapshot | pre or post publication (optionally on snapshots) |
| Systematic traditional reviewing workflow (1) | Yes, by OJS integration, a closed review phase is supported | N/A |
| Multiple reviewing rounds (1) | Yes, by OJS integration | N/A |
| User-centered publishing possibility (5) | Restricted to PDF | web-based adaptive rendering |
| Publishing reviews with articles (5) | N/A | Yes, over the Web |
| Post-publication reviews (6) | not currently supported | Yes |

*4.3. The AR-Annotator Article and Review Annotator*

As a translation between Fidus Writer/OJS and dokieli, we implemented AR-Annotator. Figure 2 shows a sample article including reviewers' comments in Fidus Writer on top, and its export to dokieli below while a user is adding a second review comment using dokieli's commenting functionality. The dokieli HTML+RDFa is generated in two high-level steps:

1. Transformation of the article's explicit structure, including metadata and comments, to dokieli's HTML+RDFa format.
2. Identification of structure (discourse elements) in the unstructured text of the article, and enrichment of the article by an explicit representation of this structure.

For example, step 1 gives us the basic structure of a section in the article, whereas step 2 marks the *subject* of that section, e.g., a section titled "Introduction" will be annotated as an instance of *deo:Introduction*, a class from the "Discourse Elements Ontology" of the SPAR family.

Three modules in AR-Annotator realize this workflow. First, a **Document Reader** reads document elements of Fidus Writer articles individually. To stay faithful to the content of the original article, it records all the article elements and their metadata. This metadata include the list of authors, table and image features, the types of textual elements, and comments. To preserve the structure of the original article, it reads the article sequentially from top to bottom, keeping the order of appeAR-Annotatorce of elements.

The second module is the **HTML Generator**. To realize this part, we analyzed the structure of articles written in dokieli and matched the semantics of each document element read from Fidus Writer to their target presentation in dokieli. To gain an equivalent HTML structure of the input article, this module once more produces HTML elements sequentially. It defines proper dokieli HTML patterns based on the metadata provided by document reader. Into these HTML patterns, we include a representation of the structural information as RDFa, representing, e.g., comments as Web Annotations.

The third module, the **RDFa Annotator**, works closely with the HTML generator of step 2. This module performs named entity recognition on common information units—including article title, abstract, section headers, author names, references, images and tables—in the article and then maps them to corresponding RDFa patterns using suitable RDF vocabularies. To create this list of keywords, called gazetteer in natural language processing, we reviewed section titles of 40 articles in the field of computer science that used a variety of terminologies and article structures, took their title sections, and mapped them to entities in our ontology. This set comprises names of entities such as "Introduction and Motivation", "Conclusion", etc. We mapped these keywords to classes defined in the *schema.org* [32], *DEO* [33] and *SWRC* [34] ontologies.

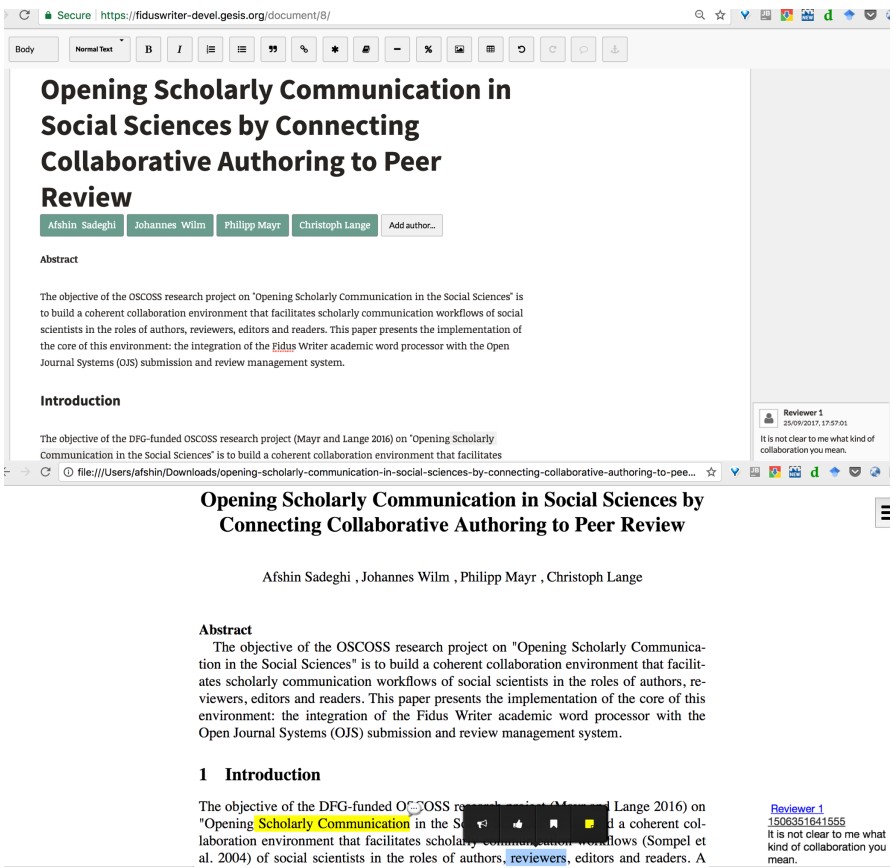

**Figure 2.** A reviewed article in Fidus Writer (**top**) and dokieli (**bottom**).

AR-Annotator finally creates a self-contained ZIP archive containing the HTML+RDFa article, separate HTML files (embedded into the article) including the review comments, as well as accompanying media files. This archive is suitable for deploying on any server and also for offline use.

For evaluation, we provide a sample article created by our approach available online [35]. We also implemented a plugin [16] to invoke our approach from Fidus Writer. Although users can edit the automatically generated annotations in the resulting dokieli document, this plugin also supports a user-friendlier manual assignment of annotations in Fidus Writer before running the export.

## 5. Usefulness of Publishing Fine-Grained Review Comments

The requirement for a system to support fine-grained reviewers' comments may not be accepted universally, and even though most state-of-the-art document authoring and viewing tools support such fine-grained annotations, as discussed in Section 1.2, peer reviewers rarely make use of them, and editors *encourage* their use even more rarely. To confirm the importance of this requirement, we have therefore conducted a preparatory study, which, to the best of our knowledge, has not been conducted before: comparing the usefulness of reviewers' comments attached to phrases inside articles in a fine-grained way to the traditional practice of publishing reviews as one long block of text.

### 5.1. Study Participants

Of the researchers attending TPDL 2017, the Conference on Theory and Practice of Digital Libraries, we invited 6 persons to participate in this survey and divided them into two groups.

### 5.2. Study Design

We first selected three reviewed articles from the Semantic Web Journal [36], an open peer review journal established in the Web community, which had both reviews with fine-grained comments attached (implemented as PDF annotations) and traditional all-in-one review texts. We then chose from the reviews 10 issues overall that referred to a section of the article and that had been pointed out both by a reviewer attaching fine-grained annotations and a reviewer using traditional techniques. We designed questions about these comments (see example in Figure 3), such that answering them would require the participants to read both the reviewer's comment and the part of the article addressed by the comment. We presented the articles and the reviewers' comments as hard copy printouts to the test subjects in two forms. In the first form, ranges of the article text were highlighted in the place that a reviewer had commented on, and the comments were displayed next to these highlights. As, so far, few reviewers make use of PDF annotations, whereas we intended to compare the usefulness of comments attached as fine-grained annotations to the usefulness of text blocks making written references to parts of an article, we paid attention to providing the test subjects with reviews that actually made use of fine-grained annotations, regardless of whether the reviewers had done so in their actual reviews. Therefore, where the reviewers had not themselves created fine-grained annotations, *we* attached them to copies of the articles, aiming at faithfully conveying the messages intended by the reviewers. The second form was the traditional display of the article without any highlights, and the reviews on a separate page. We assigned to each group of study participants one of the two formats of reviewed articles. We asked them to answer the questionnaire and we recorded the time it took them to answer the 10 questions. As an example, one of these questions is shown in Figure 3. The list of all selected articles and their reviews along with the questionnaire is available online [37].

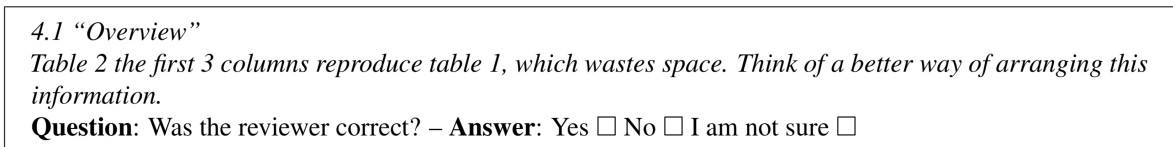

*4.1 "Overview"*
*Table 2 the first 3 columns reproduce table 1, which wastes space. Think of a better way of arranging this information.*
**Question**: Was the reviewer correct? – **Answer**: Yes ☐ No ☐ I am not sure ☐

**Figure 3.** A reviewer's comment and a question about it.

### 5.3. Evaluation Result

Comparing the time consumed by the two groups of study participants shows that reviews that are connected to the parts of the article they address in a fine-grained way are faster to consume than

separated reviews. The group that was provided fine-grained review comments finished their task in nine minutes on average, whereas the group that was provided separated reviews finished their task in 20 min on average. In addition, the participants found it more difficult to answer questions when the reviews were not attached to the affected parts of articles. These observations confirm the relevance of requiring that our proposed system should support the publication of fine-grained review comments attached to parts of articles.

## 6. Conclusions

Aiming at opening scholarly communication and specifically at opening the peer review process, we introduced a framework that allows for opening a peer review workflow that is supported by the Fidus Writer academic word processor and the OJS review management system. Once an article has been "finished", our method exports it with the reviewers' comments and feedback attached to it to the dokieli HTML+RDFa format. Technically, this process is as straightforward as implementing a variant of one of the existing Fidus Writer export filters to HTML, DOCX or Open Document, but the current export filters of Fidus Writer do not export comments. Thus, we developed an exporter that adds this feature.

Although the exporting of comments may not be novel, the combination of comments with linked data and their application in open reviewing is novel. Furthermore, our approach produces dokieli articles from articles that are made using Fidus Writer and enriches them with standard ontologies. This promotes semantically linked open data and makes an article machine readable.

As future work, we plan to extend the current semi-automated analysis setup to a system that supports automated review analysis. Such a system, besides extracting RDFa data from a paper, directly performs queries and shows the results in the article.

Secondly, we plan to apply AR-Annotator on a corpus of articles to extract RDF datasets form articles at a large scale and integrate the extracted information with the Scholarly Communication Metadata Knowledge Graph we generated previously [15].

**Supplementary Materials:** A source code to use our approach in Fidus Writer is available online [16].

**Author Contributions:** All the authors contributed in the conceptualization, methodology, writing, review & editing of the this research. The main contributor in investigation, methodology, software development, validation and the original draft is A.S., S.C. and J.W. contributed specifically in the methodology. C.L. and P.M. mainly contributed in conceptualization, writing and reviewing, supervision, validation, project administration and funding acquisition of the research.

**Funding:** This research was partially funded by German Research Foundation (DFG) under Grant No. AU 340/9-1 and SU 647/19-1.

**Acknowledgments:** This work is developed in the context of the OSCOSS (Opening Scholarly Communication in the Social Sciences) project and has been partially funded by DFG under Grant No. AU 340/9-1 and SU 647/19-1.

**Conflicts of Interest:** The authors declare no conflict of interest.

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
