# Peer review of "Opening and Reusing Transparent Peer Reviews with Automatic Article Annotation"

_publications, doi:10.3390/publications7010013_

Round 1
Reviewer 1 Report
Excellent paper and thank you for submitting it to us
Author Response
Thank you for your effort and the help in increasing the scholarly communication quality.
Reviewer 2 Report
This paper builds a tool/workflow for open reviewing.
I also find open reviewing a positive change to reviewing process, the biggest challenge is to change the community attitude about adopting it, not any missing tools. Therefore, any tools will not be really useful, and certainly need not provide a database or the reviews, to answer queries!
Therefore, the presenter requirements seem unrealistic.
The examples described [111-113] are trivial.
Why should only reviewers (and not all users) be given access to underlying resources [127-128]? How would then the users judge the reviewer comments?
Requirement lbbel [135-137] should take into account other common means of comments and annotations, e.g. ms-word.
Requirement ldbel [140-142] would be nice to do for all documents/articles, even without considering the reviews. It should be examined in a different context.
I dont think that requirements lebel and lfbel [143-145] will really have any use - from common readers/reviewers.
What does the CL@AS and AS@CL [156, 158] mean?
A survey with 6 persons on 3 articles and 10 comments is too small, and any results are not trusted.
The difference in finishing times [306-312] is biased, as (a) the "annotated accordingly articles" [300] is a type of preprocessing that works in favour of the one team only and (b) the task itself need not be performed by most users: how often did real users need to identify and read the specific parts of the article that the reviews refer to (outside of this study - and excluding the article authors)? I believe that in most cases they do not need such fine grained detail.
I find that the work done (e.g. the comment exporter) are not such useful or novel.
Author Response
Response to Reviewer 2 Comments Thank you for your effort and the help in increasing the scholarly communication quality. Point 1: I also find open reviewing a positive change to reviewing process, the biggest challenge is to change the community attitude about adopting it, not any missing tools. Therefore, any tools will not be really useful, and certainly need not provide a database or the reviews, to answer queries! Therefore, the presenter requirements seem unrealistic. Response 1: Hereby we try to sharpen a bit our argumentation on how our tool is supporting the change of mind in the community. Our argumentation is that when a task is easier it is more probable that people try that. Besides we show to users that there are many benefits in choosing this type of reviewing. Point 2:The examples described [111-113] are trivial. Response 2: We updated them with new query examples. Point 3: Why should only reviewers (and not all users) be given access to underlying resources [127-128]? How would then the users judge the reviewer comments? Response 3: We removed the word "reviewers" from the sentence and made the requirement more general. Point 4: Requirement lbbel [135-137] should take into account other common means of comments and annotations, e.g. ms-word. Word is a tool that (partly) implements this requirement. In Word, some one can attach fine-grained comments to a document. Just the publishing is a bit limited: Some one could only upload the Word file, or a PDF export (including comments) to the Web. Therefore, we consider this requirement still valid. Response 4: In Word, although it is possible to attach fine-grained comments to a document, the publishing is a bit limited: you could upload the Word file, or a PDF export (including comments) to the Web. Point 5:Requirement ldbel [140-142] would be nice to do for all documents/articles, even without considering the reviews. It should be examined in a different context. Response 5: Added a note [in the line 150] that this requirement applies to the publishing of articles in general, not just to peer-reviewed articles. Point 6:I dont think that requirements lebel and lfbel [143-145] will really have any use - from common readers/reviewers. Response 6: We pointed out the FAIR principles [in lines 144-146] to emphasize the relevance of requirement 5; linked the criteria in the feature comparison table explicitly to requirements. Point 7:What does the CL@AS and AS@CL [156, 158] mean? Response 7: It was a latex error, it is removed. Point 8: A survey with 6 persons on 3 articles and 10 comments is too small, and any results are not trusted. Response 8: We understand that although “the 5 users is enough” argument [https://measuringu.com/five-users/ https://dl.acm.org/citation.cfm?id=169166] argument applies to certain qualitative questions of usability, we tried to design the study such that follow that. It was also very difficult to gather enough experts with experience of reviewing in one topic in one place. Point 9:The difference in finishing times [306-312] is biased, as (a) the "annotated accordingly articles" [300] is a type of preprocessing that works in favour of the one team only and (b) the task itself need not be performed by most users: how often did real users need to identify and read the specific parts of the article that the reviews refer to (outside of this study - and excluding the article authors)? I believe that in most cases they do not need such fine grained detail. Response 9: We have clarified the presentation of the experiment. We explained [in lines 304-308] that the purpose was to compare fine-grained annotations vs. old-school text blocks, rather than comparing actually existing reviews containing some fine-grained annotations to actually existing reviews not containing any. Point 10: I find that the work done (e.g. the comment exporter) are not such useful or novel. Response 10: Although the exporting of comments may not be novel, the combination of comments with linked data and their application in open reviewing is novel. Furthermore, our approach produces dokieli articles from articles that are made using Fidus Writer and enriches them with standard ontologies. This promotes semantically linked open data and makes an article machine readable. [mentioned in lines 331-333]

Reviewer 3 Report
The technological solution is interesting and the overall article is well written.
The Paper's title should be revised. The word sequence is not written in a plain English. The title does not include the ideia of a technological proposal. I suggest changing the sentence format in this way: Opening and Reusing (Transparent?) Peer Reviews with Articles' Automatic Annotation.
The Introduction should include some essential concepts such as academic publishing or schorlarly communication and highlight the connection between open peer review (and its benefits) and open science.
The introduction should also be redesigned to include section 0.3 in the first part of introduction, avoiding sentences like «The remainder of this article is structured as follows».
Author Response
Firstly, Thank you for your effort and the help in increasing the scholarly communication quality.
Point1: The Paper's title should be revised. The word sequence is not written in a plain English. The title does not include the idea of a technological proposal. I suggest changing the sentence format in this way: Opening and Reusing (Transparent?) Peer Reviews with Articles' Automatic Annotation.
Response 1: The Title is updated accordingly.
Point 2: The Introduction should include some essential concepts such as academic publishing or scholarly communication and highlight the connection between open peer review (and its benefits) and open science.
Response 2: We added a sentence (lines 48-50) to introduction adding two essential concepts with 3 new references.
Point 3: The introduction should also be redesigned to include section 0.3 in the first part of introduction, avoiding sentences like «The remainder of this article is structured as follows».
Response 3: the Introduction (in lines 25-35) is updated based on the suggestion and section 0.3 is moved to the first part of the Introduction. The sentences like «The remainder of this article is structured as follows» are avoided as well.

Round 2
Reviewer 2 Report
Good
Reviewer 3 Report
I have no present issues regarding this paper.